# Mapping residual transmission for malaria elimination

Robert C Reiner Jr[1,2]*, Arnaud Le Menach[3], Simon Kunene[4],
Nyasatu Ntshalintshali[3], Michelle S Hsiang[5,6,7], T Alex Perkins[1,8,9],
Bryan Greenhouse[10], Andrew J Tatem[1,11], Justin M Cohen[3], David L Smith[1,12,13,14]

[1]Fogarty International Center, National Institutes of Health, Bethesda, United States; [2]Department of Epidemiology and Biostatistics, Indiana University School of Public Health, Bloomington, United States; [3]Clinton Health Access Initiative, Boston, United States; [4]National Malaria Control Program, Manzini, Swaziland; [5]Department of Pediatrics, University of Texas Southwestern Medical Center, Dallas, United States; [6]Malaria Elimination Initiative, Global Health Group, University of California, San Francisco, San Francisco, United States; [7]Department of Pediatrics, University of California, San Francisco Benioff Children's Hospital, United States; [8]Eck Institute for Global Health, University of Notre Dame, Notre Dame, United States; [9]Department of Biological Sciences, University of Notre Dame, Notre Dame, United States; [10]Department of Medicine, University of California, San Francisco, San Francisco, United States; [11]Department of Geography and Environment, University of Southampton, Southampton, United Kingdom; [12]Spatial Ecology and Epidemiology Group, Department of Zoology, University of Oxford, Oxford, United Kingdom; [13]Institute for Health Metrics and Evaluation, University of Washington, Seattle, Washington, United States; [14]Sanaria Institute for Global Health and Tropical Medicine, Rockville, Maryland, United States

*For correspondence: rcreiner@ indiana.edu

Competing interests: The authors declare that no competing interests exist.

**Abstract** Eliminating malaria from a defined region involves draining the endemic parasite reservoir and minimizing local malaria transmission around imported malaria infections. In the last phases of malaria elimination, as universal interventions reap diminishing marginal returns, national resources must become increasingly devoted to identifying where residual transmission is occurring. The needs for accurate measures of progress and practical advice about how to allocate scarce resources require new analytical methods to quantify fine-grained heterogeneity in malaria risk. Using routine national surveillance data from Swaziland (a sub-Saharan country on the verge of elimination), we estimated individual reproductive numbers. Fine-grained maps of reproductive numbers and local malaria importation rates were combined to show 'malariogenic potential', a first for malaria elimination. As countries approach elimination, these individual-based measures of transmission risk provide meaningful metrics for planning programmatic responses and prioritizing areas where interventions will contribute most to malaria elimination.

## Introduction

Malaria is a leading cause of childhood morbidity and mortality (**Naghavi, 2015**), and chronic malaria infections contribute to and complicate diagnosis and treatment of other diseases. Controlling malaria to reduce its burden has long been a top global health priority, and eradication became an official policy of the United Nations in 1955 (**Mayo and Brady, 1955**). The ensuing Global Malaria Eradication Campaign (GMEP, 1955–1969), directed by the World Health Organization, contributed

**eLife digest** Swaziland has set a national goal of eliminating malaria transmission in the very short term, which would make it the first country in sub-Saharan Africa to do so. More than half of the cases of malaria that are observed in Swaziland are caused by infections picked up by travelers while they were in other countries where the disease is much more prevalent. The other cases – people who became infected in Swaziland – are the cases that the government of Swaziland is trying to prevent.

If Swaziland is going to eliminate malaria, it will need to identify any places where the malaria parasites are still spreading throughout the population so it can target those communities with effective prevention measures. It will also need to manage the risk that infections imported from abroad may re-start transmission in places where it has been stopped.

To work out how likely it is that a malaria infection will be transmitted by mosquitoes in a particular place, researchers can look at past malaria data and calculate how many new infections are caused by each case. Reiner et al. have now produced a computer model that estimates how this number varies across Swaziland, highlighting places where the government is going to need to focus efforts to eliminate malaria. The model shows that in some rural areas near Mozambique, each individual infected with malaria is causing more than one other person to become infected. This confirms that the disease has not yet been eliminated from these areas. However, in other regions of the country, malaria rarely spreads between individuals.

The detailed regional information from the model may help public health authorities in Swaziland better target their anti-malaria resources. In large cities where most cases are imported, Reiner et al. suggest focusing resources on providing preventive treatment to travelers who plan on visiting places where malaria is spreading. However, in rural areas where malaria continues to spread, preventively treating the whole population or providing them with tools to protect them from mosquitoes might be more appropriate. Similar considerations of regional differences in the spread of malaria could also help other countries to more effectively combat the disease.

to a massive, permanent contraction in the geographic range of malaria (*Gramiccia et al., 1988*; *Smith, 2013*), and it led to development of concepts and methods for eliminating malaria that remain useful today. A key insight was the need to use different metrics, suited to natural constraints and programmatic needs, to mark progression from endemic malaria to elimination (*Hay et al., 2008*). Malaria eradication is once again recognized as a global priority, and as of 2011, 36 of 99 malaria endemic countries had plans to eliminate malaria (*Cotter, 2011*).

Concepts and methodologies developed for the GMEP are being revisited and updated today in light of contemporary challenges and new technologies. Three useful concepts were called *vulnerability* (i.e., the rate of malaria importation), *receptivity* (i.e., the potential for ongoing local transmission), and their product, which was called *malariogenic potential* (i.e., the expected number of cases that could occur as a result of vulnerability and receptivity) (*Pampana, 1969*). Given that GMEP was predicated on the idea of universal insecticide spraying throughout at-risk areas, these notions were used primarily to evaluate the vigilance required to prevent reestablishment of transmission post-elimination rather than to make decisions about intervention selection or targeting (*World Health Organization, 2012*). They thus require updating in light of the different assumptions of the present campaigns. Malaria importation rates for African malaria elimination countries are much higher than those for countries that eliminated malaria in past decades due to high mobility and highly endemic neighbors (*Tatem, 2010*), and receptivity may be greater. Countries have multiple tools at their disposal but must determine how to deploy them optimally to achieve and sustain elimination given constrained resources. These challenges are now being faced by Swaziland, a sub-Saharan country that has made substantial progress towards eliminating malaria. Success would make it the first mainland sub-Saharan country to eliminate malaria, and so the lessons from its national experience will be relevant for the rest of the continent.

## Residual transmission and elimination

A key need is for new quantitative methods and practical operational advice to guide the final stages of malaria elimination, when few local cases remain. Elimination programs have repeatedly demonstrated the critical importance of identifying areas where transmission continues in order to make the most of limited resources. The architects of the smallpox eradication strategy, for example, credit the campaign's ultimate success to a shift from universal to targeted vaccination (*Foege et al., 1971*). Successful elimination of malaria during the GMEP similarly demonstrated that, in resource-constrained environments, a shift is required away from a focus on universal coverage for endemic malaria towards heightened surveillance, case investigation, identification of areas where transmission risk remains high (i.e., residual transmission foci), and highly targeted interventions (*Moonen, 2010*). Despite progress (*Ja and Hg, 2009*), little guidance is available on what specific methods to use where and which metrics are appropriate for measuring progress in today's eliminating countries.

Since 1999, Swaziland's National Malaria Control Program reports that the incidence of malaria has declined from 2.9 to 0.07 malaria cases per 1,000 people per year (*Roll Back Malaria, 2012*). Between 2010 and June 2014, Swaziland confirmed only 2,129 malaria cases, with case investigation of 1,524 of them. Of these, 870 (57%) were classified as imported, with the proportion of cases likely having acquired their infection elsewhere increasing since 2010. At a national level, the decreasing ratio of local to imported malaria cases since 2010 is suggestive of an reproductive number under control, $R_c$, much less than 1 on average, which would mean that elimination of endemic transmission may have already been achieved or is imminent (*Churcher et al., 2014*).

Analysis of national trends in reporting data provides a useful measure of overall progress but potentially masks local spatial heterogeneity in transmission, and it leaves unanswered the question of how to stratify Swaziland to allocate resources most efficiently within the country. Ideally,

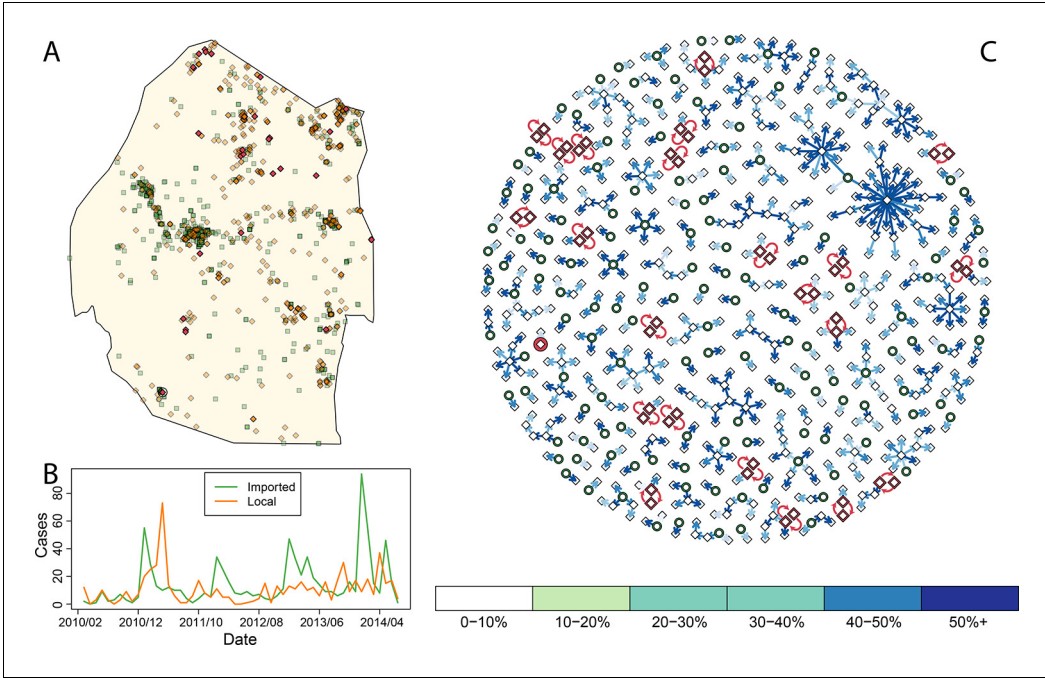

**Figure 1.** Consensus network plot of causal links. Panel A: Swaziland imported and local malaria cases (green squares and orange diamonds, respectively) are plotted spatially. Local case pairs identified as putative orphaned chains are indicated by red diamonds. A solitary local case also identified as an orphan is identified as a red diamond within a circle. Panel B: Swaziland imported (green line) and local (orange line) malaria cases are plotted in time, aggregated by month. Panel C: The final consensus network plot is displayed. Local cases are plotted as diamonds and imported cases as green circles. The color of each link corresponds to the "strength" of the connection as measured by the number of parameter sets where that link was identified as optimal. Imported cases that were not found to be the "most likely" parent of a local case are not displayed.

programs would focus attention on places and at times where and when the risk of malaria transmission is highest. Programs might direct aggressive interventions, such as focal mass drug administration, towards residual endemic foci where transmission would persist even in the absence of importation, but might opt for less aggressive maintenance of reduced risk in places where transmission is driven only by continual replenishment from imported infections. Determining whether locally acquired malaria infections are the result of endemic transmission or result from transmission chains stemming from importation requires analyzing transmission at an individual level. National risk maps have been developed based on the household locations of local malaria cases (*Cohen, 2013*) but these maps do not describe how much transmission is likely occurring, only whether or not there is risk of any local infections in a given location as narrowly defined by the location of infected individuals. Furthermore, they do not explain the relative importance of importation as a driver of transmission and thus cannot inform intervention selection.

Improving assessment of progress and making action maps requires developing individual-based assessment of risk to link cases together, determine the magnitude of transmission as measured by $R_C$, and evaluate where transmission may be occurring endemically versus where it is driven only by ongoing importation. Most of the imported cases have been identified in and around the large cities of Mbabane and Manzini (*Figure 1A*), but proportionally fewer locally acquired cases were found in these cities (*Cohen, 2013*). Malaria cases imported into the major cities of Swaziland are likely only responsible for causing a few local cases (i.e., $R_c$ in the large cities is very low). Areas outside the major cities—specifically those in the east and north of Swaziland—appear to have a higher ratio of local to imported cases, suggesting that though on average each malaria case generates less than one other case (i.e. $R_c < 1$), there are some focal areas where endemic transmission may continue to smolder. Measuring progress and achieving elimination requires properly characterizing and quantifying heterogeneity in this residual transmission. Here, to address these needs, we have defined *vulnerability*, *receptivity*, and *malariogenic potential* in quantitative terms based on a continuous point-process and developed fine-grained maps.

## Mapping receptivity

Elimination programs in the endgame have long been advised to categorize cases as imported, introduced (the result of first degree transmission from an imported infection), or indigenous (the result of second or more degree transmission), yet no method for making this distinction has ever been formally described (*Pampana, 1969*). Methods for doing so are largely based on case investigations, but such methods could be validated and augmented with genetic and computational methods for linking cases. Understanding which cases are linked to other cases is also important because it allows direct measurement of $R_C$, an important measure of the need for additional intervention. We developed computational methods for assessing malaria transmission (see Methods) that quantifies the most likely "parent" of each local case. We do not distinguish between potential parents in terms of their status as "imported" or "local". In this, we are estimating causal links that can be counted in space-time to get measurements of receptivity. Further, these links build transmission chains that can be used to gain a deeper understanding of the spatial variation in $R_c$. This approach uses an understanding of malaria transmission dynamics to redefine "proximity" of two malaria cases as a generalized probabilistic measure based on distance and time separating two cases to evaluate potential causality. Malaria transmission directly links two individuals through two bites from the same mosquito. The time elapsed between detection of two linked cases will be bound by the serial interval (*Fine, 2003*) (i.e., the length of a complete malaria transmission cycle), mosquito mortality, and the timing of case detection relative to infection. These aspects of malaria ecology provide probabilistic bookends to temporal proximity, as even with detection (and assuming all detected cases are treated promptly), cases are ever more unlikely to be linked as the chance of a mosquito living long enough to link them diminishes. In space, the simple but common Gaussian diffusion-based approximation of movement approximates the combined spread of mosquitoes and humans, and is governed by a single constant. Given the use of continuous distribution functions, the algorithm was forced to always find a link even if it was inconceivable. As such, we incorporated a minimum threshold that dichotomized links into "plausible" (with a level of plausibility) and "implausible."

By accounting for potential differences in the distribution of the serial interval for human malaria cases and timing of case identification, multiplying the temporal processes with spatial diffusion, and

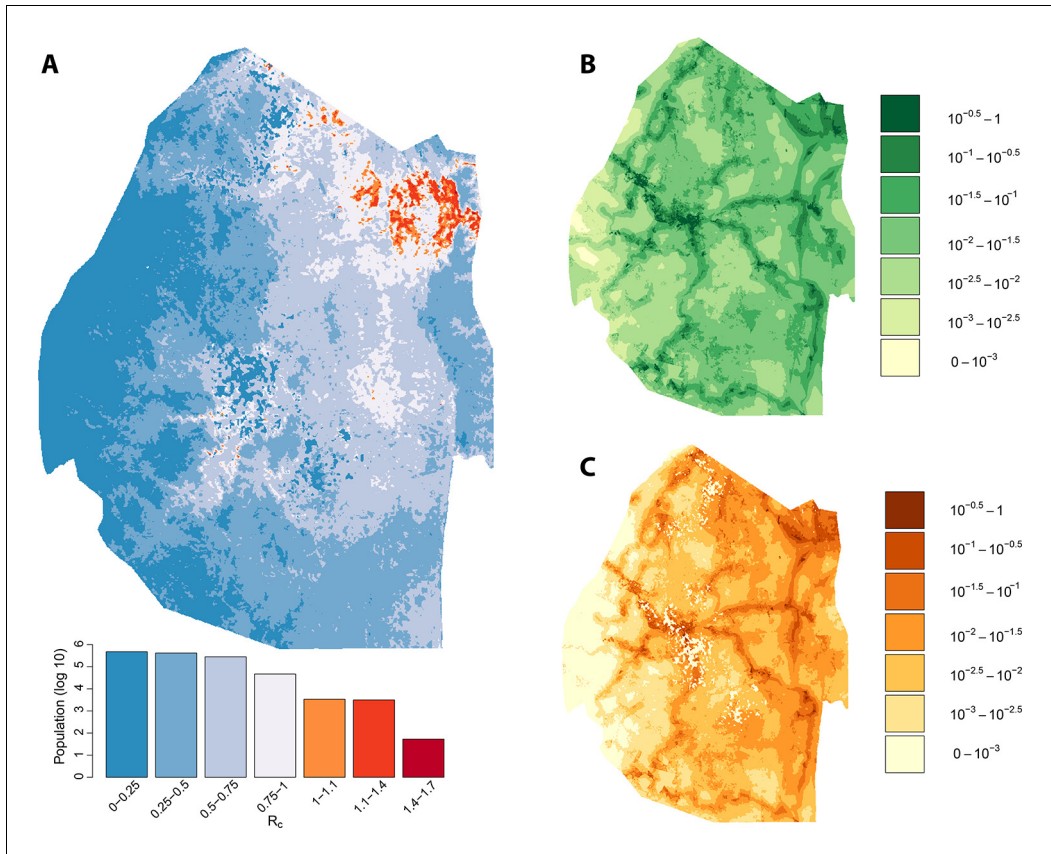

**Figure 2.** Vulnerability, receptivity and malariogenic potential. Panel A: Extrapolated $R_c$ values for Swaziland using a zero-inflated negative binomial regression. Areas in orange to red indicate locations where $R_c$ is greater than unity. The legend doubles as a histogram indicating the number of individuals (on a $\log_{10}$ scale) that live within each range of $R_c$ values. Panel B: Extrapolated importation probabilities for Swaziland using a logistic regression. Panel C: Malariogenic potential for Swaziland calculated as the product of $R_c$ and the probability of importation.

finally sweeping across a suite of potential diffusion constants (see Methods) we linked local cases to a "most likely" parent case. By combining "most likely" links, we arrived at a single weighted network that represents the consensus linkages (*Figure 1*). As was suspected by visual inspection of the spatial distribution of the data, most of the imported cases within the major cities of Swaziland did not appear to be responsible for ongoing transmission. However, both imported and locally transmitted cases in the north and east did, infrequently, initiate extended transmission chains. Elsewhere, no identified case was a likely parent, so these cases are classified as "orphans."

Taking the output of the transmission network, the number of direct 'offspring' arising from each case is its estimated $R_c$. Using zero-inflated negative binomial regression models on a set of ecological, social and demographic covariates, likely values of $R_c$ were extrapolated spatially from observed case locations to all of Swaziland at 100-meters squared resolution (*Figure 2A*). The resulting map of $R_c$ illustrates the estimated heterogeneous distribution of current malaria receptivity within Swaziland. To the west, $R_c$ is close to 0. Within Mbabane and Manzini, $R_c$ is estimated at 0.08 and 0.12 respectively. Conversely, in the northeast near the Mozambican border, $R_c$ estimates increase up to 1.70. As suspected, while endemic transmission does not appear to occur within the urban centers of Swaziland, it does not yet appear eliminated from the entirety of the country. The smoothed $R_c$ map was found to be statistically significantly different from a flat map (p-value <2e-16) but it remained unclear the statistical significance of each pixel. Further analysis is required to assess the robustness of isolated locations where $R_c$ appears elevated. This will be an important step to appropriately interpret these maps for the purpose of resource allocation given constraints on the number of individual locations that can be visited.

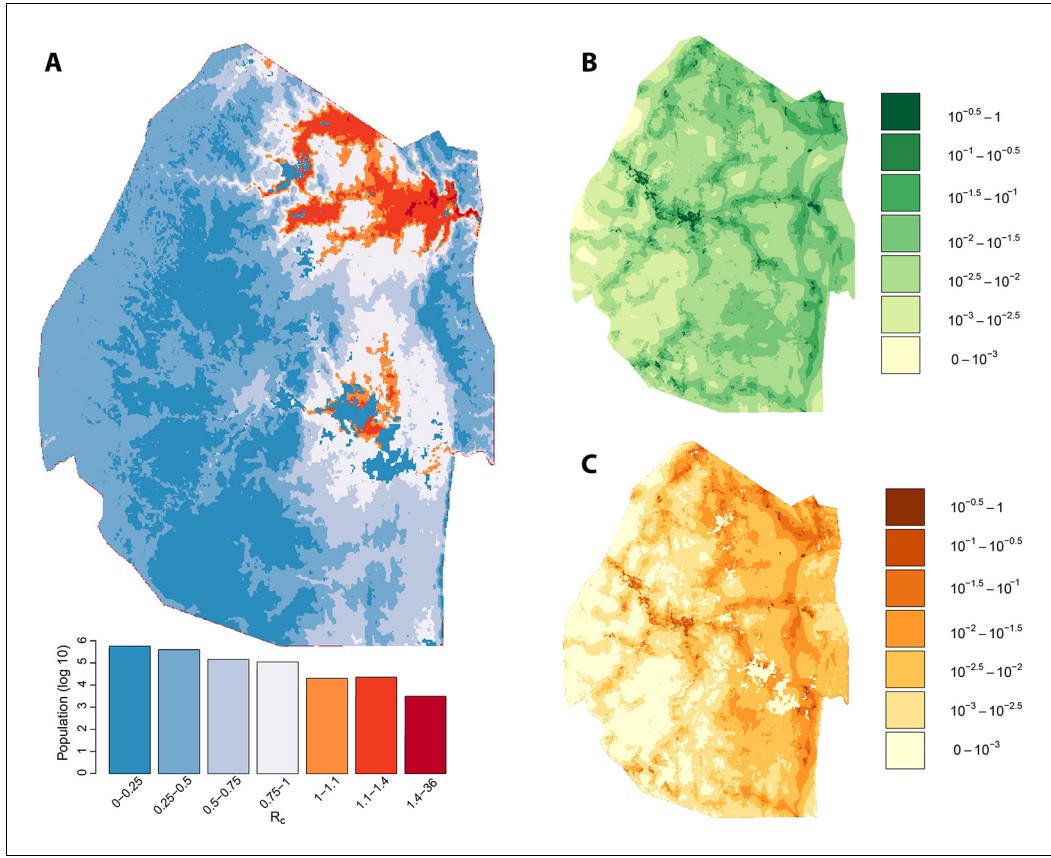

**Figure 3.** Vulnerability, receptivity and malariogenic potential (2010-6/2012). Panel A: Extrapolated $R_c$ values for Swaziland using a zero-inflated negative binomial regression. Areas in orange to red indicate locations where $R_c$ is greater than unity. The legend doubles as a histogram indicating the number of individuals (on a $\log_{10}$ scale) that live within each range of $R_c$ values. Panel B: Extrapolated importation probabilities for Swaziland using a logistic regression. Panel C: Malariogenic potential for Swaziland calculated as the product of $R_c$ and the probability of importation.

## Malariogenic potential

The potential for local transmission will not result in actual transmission unless malaria parasites are present. As Swaziland successfully extirpates these final foci of endemic transmission, local transmission in the country will increasingly arise only around imported malaria cases. The total number of locally acquired malaria cases in Swaziland is the product of importation (vulnerability) and onward transmission (receptivity), with the ongoing operational challenge of maintaining gains greatest in regions with both higher receptivity (i.e. $R_C$) and higher vulnerability (i.e., the number of malaria infections imported each year that could seed new transmission): the product of these quantities has been defined as the 'malariogenic potential' (*Pampana, 1969*). Malariogenic potential was mapped as the product of $R_c$ and vulnerability (*Figure 2c*). The resulting map illustrates where locally acquired/transmitted cases are most likely to occur, and thus where resources may need to be prioritized to prevent reestablishment of malaria given the joint risks of importation and subsequent transmission.

We assessed the stability of our malariogenic potential maps as well as our other output by splitting the data into two halves (before and after July 1, 2012). The malariogenic potential maps for the two halves appear very similar (*Figure 3C* versus *Figure 4C*), as are the importation maps (*Figure 3B* versus *4B*). There are some differences in the $R_c$ maps (*Figure 3A* versus *4A*), but both analyses identified regions in the northeast where $R_c$ was larger than 1. Both analyses identified a larger 'max' $R_c$ (35.36 for the first half and 3.13 for the second half) as well as a larger percent of the population living in the highest $R_c$ category ($R_c > 1.4$).

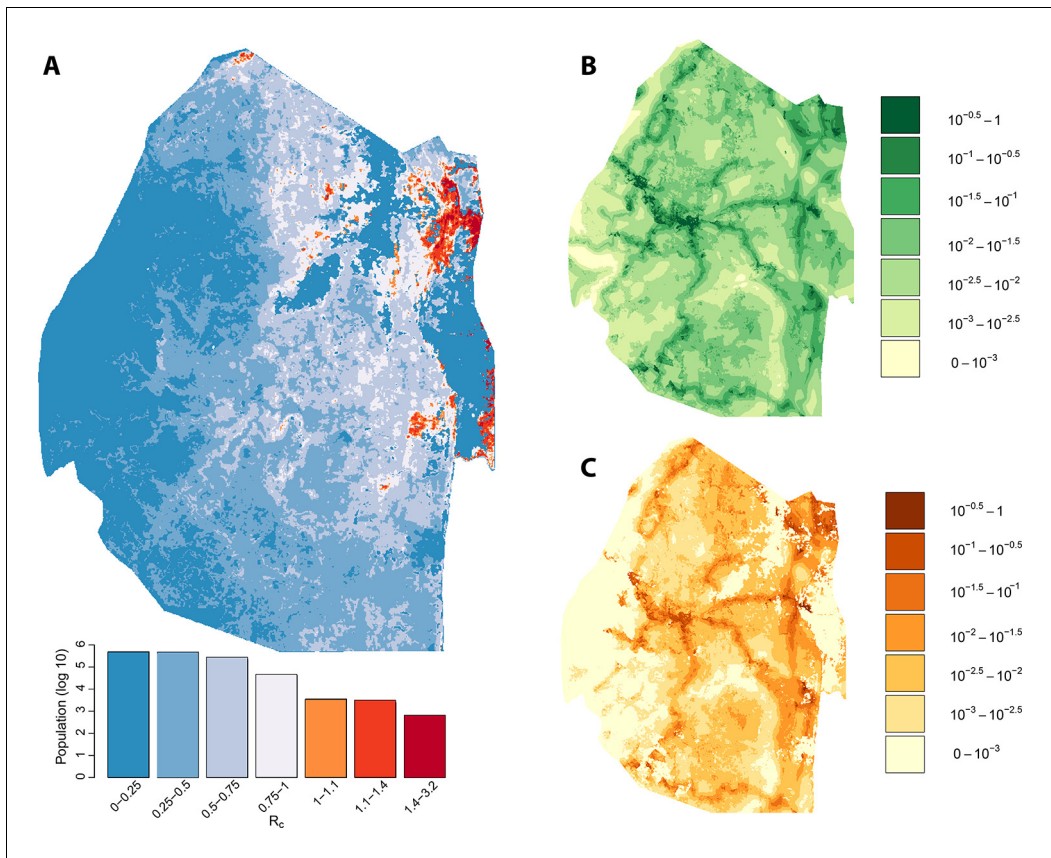

**Figure 4.** Vulnerability, receptivity and malariogenic potential (7/2012-2014). Panel A: Extrapolated $R_c$ values for Swaziland using a zero-inflated negative binomial regression. Areas in orange to red indicate locations where $R_c$ is greater than unity. The legend doubles as a histogram indicating the number of individuals (on a $\log_{10}$ scale) that live within each range of $R_c$ values. Panel B: Extrapolated importation probabilities for Swaziland using a logistic regression. Panel C: Malariogenic potential for Swaziland calculated as the product of $R_c$ and the probability of importation.

## Occult local transmission

Ideally, every case of malaria in Swaziland would be detected. In reality, case detection is imperfect, and it is likely unnecessary to find every case to create circumstances that lead to elimination. In this analysis, missed cases could matter if they biased the estimates of $R_c$, depending on whether locally transmitted or imported cases were more likely to be missed. If a case that is missed results in local transmission and if those future cases are captured by surveillance, those future cases may appear to have no plausible cause. The identification of such "orphan" cases can help indicate places where Swaziland must work to implement or strengthen active infection detection. Within this analysis, due to the inclusion of temporal uncertainty to account for potential differences between detection times and time of onset, rare links can be formed between a case and a second case where the second case was picked up at the same time or even later. These rare links would only form if no other case that occurred earlier were close in space-time as judged by the spatio-temporal kernels. In these circumstances, due to the flexibility of the spatio-temporal kernel, a pair of cases could be identified as being the most likely parent of each other. These "loops" identify two orphaned cases that were identified close in time-space to each other but whose actual parent was not captured by surveillance.

Within this analysis, there were 22 pairs of local cases that formed loops (*Figure 1A*, *Figure 1C*, red diamonds) as well as a single case that was not linked to any other case across any of the potential parameter sets (*Figure 1A*, circled red diamond, *Figure 1C*, red circle). Prioritizing enhanced

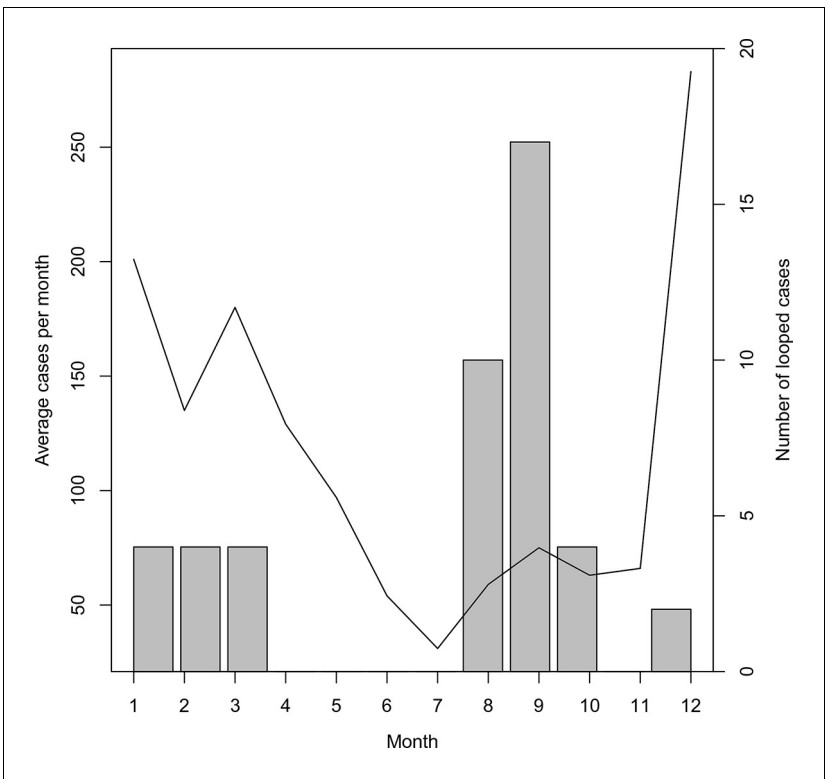

**Figure 5.** Timing of 'orphan' cases. The average number of cases per month and total occurrence of looped (or 'orphaned' cases) are plotted against month.

surveillance in the areas surrounding these orphan chains would help narrow uncertainty about residual transmission within Swaziland.

The majority of these loops occurred during the months with the least transmission (*Figure 5*), which reflects the decreased chance of any cases being detected in the month prior. The method described here yields estimates of transmissibility that can guide interventions to places where occult transmission is most likely to be happening even in the absence of knowledge of specific infected individuals. Nevertheless, the more complete the surveillance effort, the more accurate these mapping efforts will become.

Asymptomatic infections are another possible explanation for the orphaned cases. Missing asymptomatic cases could either result in an overestimate or underestimate of $R_c$ (if they are likely parents or likely offspring of other cases). For this analysis, we did not account for asymptomatic or inapparent cases. Although it was not done universally, intensive infection detection around cases resulted in very few additional infections (53/7307 between July 2014 and June 2015) consistent with the assumption that there is not a large pool of unreported infections that would greatly bias our results.

## Discussion

Communicable disease policies require different approaches than policies for non-communicable diseases, as each case presents both medical and public health challenges (*Smith et al., 2005*). For infectious diseases, reproductive numbers provide a theoretical basis for strategic planning and programmatic evaluation, such as critical vaccine coverage levels and outbreak responses (*Anderson and May, 1991*). Estimating individual reproductive numbers by linking up malaria infections is a special case of a method that has be used more widely for other diseases (*Walker et al., 2010*).

Malaria, like other diseases with an environmental component, represents a special challenge because of spatial heterogeneity in the risk of transmission (*Bejon et al., 2010*; *Bousema et al.,*

*2012*; *Bejon et al., 2014*; *Bousema et al., 2010*). In the end phases of elimination, population-level measures become inefficient and inadequate, so as countries approach the goal of eliminating malaria, individual-based estimates of transmission must identify putative foci where transmission remains high and where resources should be targeted (*Hay et al., 2008*). Case counts alone do not necessarily convey information about transmission, since many of those cases may have been acquired far from where they were detected. Aggregate ratios of local to imported cases in time (or in space) alone, while more representative of overall progress, could obscure localized transmission if, for example, most cases failed to transmit but some pockets of transmission remained. This analysis, which identifies places and times where cases are most likely to be transmitted, confirms that there has been dramatic progress towards elimination overall, but it also identified substantial heterogeneity in progress within Swaziland.

The Swaziland National Malaria Control Program (NMCP) will need to manage imported malaria as long as endemic transmission continues in neighboring countries, so directing and optimizing limited resources is crucial. Combining assessments of receptivity with assessments of vulnerability provide actionable intelligence to support malaria programs in designing targeted intervention strategies in the most relevant places; for example, the NMCP may consider targeting travelers with prophylaxis in places with high vulnerability, while focal mass drug administration or other aggressive measures might be most appropriate in places with evidence of endemic transmission and low vulnerability. Our approach provides spatiotemporally relevant and resolved metrics of transmission that can be used to identify future cases as either critical or relatively unimportant for overall elimination efforts. Further, and perhaps most important, our approach can be used to stratify future control responses by differentiating between locations where elimination would be a consequence of merely decreasing effective importation versus where elimination of endemic transmission is needed through reduction in local receptivity. Predictions generated by our approach will also be useful as a baseline for in-development genetic testing and molecular typing models (*Greenhouse and Smith, 2015*), and will remain pertinent as a proxy for such methods in places where resources are limited to enable universal parasite typing. These methods can help Swaziland reassess its needs and remain malaria-free as surrounding countries control transmission and make further progress. Through regional elimination, economic growth, and efficient use of existing resources, malaria elimination can perhaps become as stable in Swaziland as it has been elsewhere (*Smith, 2013*; *Chiyaka et al., 2013*).

## Methods

### Surveillance data

Swaziland implemented the first stage of a national malaria elimination policy in 2011, and local malaria transmission dropped to extremely low levels. There was a 25-fold decline in the average malaria case-load, from 10.0 cases per 1000 in 1995 at the peak of an epidemic to 0.4 cases per 1000 in 2010 (*Roll Back Malaria, 2012*). Swaziland also benefitted from a regional malaria control effort called the Lubombo Spatial Development Initiative (LSDI) established as a partnership with neighboring countries. The LSDI sharply reduced the number of malaria cases imported from its neighbors, and now most of the remaining cases appear to originate in Mozambique.

At low transmission intensity, the methods and metrics used elsewhere in Africa to assess the risk of further local transmission initiated from imported malaria become inadequate, so Swaziland adopted a surveillance system based on some combination of passive and reactive case detection. Household locations of confirmed malaria cases were identified by passive or reactive case detection and georeferenced by the NMCP. Infected individuals reporting no travel, whether abroad or within Swaziland were categorized as locally acquired cases. Infected individuals who reported travel abroad to endemic countries within biologically meaningful windows were categorized as imported cases. Initially, a travel history in the previous two weeks was collected, this was updated to four weeks in August 2012 and to eight weeks in July 2013.

From January 2010 to June 2014, Swaziland investigated 1,524 cases collecting information about household location, case demographics (age, gender, occupation), use of malaria prevention measures, dates of symptoms and of diagnosis, treatment. Of all investigated cases, 592 (38.8%) were categorized as locally-acquired based on a lack of a recent travel history to endemic regions. This

national average of 592 local cases to 870 imported cases suggests that on average, the reproductive number under present levels of control is approximately 0.4. This estimate represents a national average figure, however, and it could disguise undetected ongoing transmission in some hotspots.

## Spatio-temporal associations

What is needed is a tool that can simultaneously assess transmission dynamics in a low-transmission setting such as Swaziland, identify locations that systematically produce unobserved cases, and provide internal feedback to improve the surveillance system. As a first step towards accomplishing this larger purpose, we developed algorithms to reconstruct putative transmission chains. Using comprehensive case data from the Swaziland NMCP from 2010 through 2013, these estimated chains—based on identifying likely causal links between successive cases through the use of spatio-temporal kernels—provide insight on the frequency and length of chains of local transmission.

To evaluate the relative chance that one locally acquired case arose from any other case, we would optimally like to calculate the probability that an older case was fed on and initiated a transmission cycle in a mosquito that subsequently fed on and infected the second case. This measure of propensity would combine the epidemiology of malaria within a mosquito, the mosquito's lifespan as well as movement probabilities for both mosquitoes and humans. Given the complexities of both human and mosquito movement, we assessed the likelihood using a family of probability distribution functions. By varying the unknown space and time parameters for each component, we can produce a putative single space-time distribution of locally acquired cases based on the time and location of any other case.

The approximate likelihood will be a product of mosquito lifespan, mosquito movement, human movement, and malaria epidemiology. We assume that the contributions of movement act independently of the contributions of mosquito and disease ecology. Following previous approaches, we approximate movement with simple diffusion. This movement kernel aggregates both the movement of the mosquito as well as the movement of the individuals. We assess the importance of this assumption by substituting diffusion within our algorithm with a long-tailed Pareto distribution, discussed in Appendix 2. Simple diffusion assumes that there is no trend in time (i.e. the most likely outcome is that there is no movement), and that movement is Gaussian. Specifically, if we have two individuals who became infectious $t$ days apart and $x$ meters apart, the chance, denoted $M(x,t)$, that the pathogen had *moved so far* in such a time given a specified diffusion constant, $D>0$, would be

$$M(x,t) = \frac{1}{\sqrt{2\pi Dt}} e^{-\frac{x^2}{2Dt}}$$

The mosquito and transmission dynamics component is more complex. Due to the disease ecology of malaria, incubation periods within both the mosquito (known as the *extrinsic incubation period*) and the second host (known as the *intrinsic incubation period*) definitionally separate onset of infectiousness in secondary infections from the onset of infectiousness in the initial case (the time between the onset of infectiousness in causally linked infections is known as the *serial interval*). The serial interval is the sum of the extrinsic incubation period, the time elapsed while infectious mosquitoes quest for blood and infect humans and the intrinsic incubation period. We estimate the intrinsic incubation period as minimally 18 days (approximately six days in the liver plus 12 days for mature gametocytes to be produced in sufficient densities) and the extrinsic incubation period as minimally 12 days.

The brief lifespan of the mosquito acts as an opposing force to the extrinsic incubation period. One increases the serial interval while the other decreases the likelihood of extended time between subsequent cases. Estimates of *Anopheles* mosquitos' lifespan is between 10 and 14 days. Though somewhat simplistic, the exponential distribution is frequently used for the lifespan of mosquitoes. Combining mosquito and disease ecology, if we have two individuals who became infectious $t$ days apart and $x$ meters apart, the chance, denoted $G(x,t)$, that the serial interval *was so long* would be

$$G(x,t) = \begin{cases} 0, & \text{if } t<30 \\ \frac{1}{12} e^{1-\frac{1}{12}(t-18)}, & o.w. \end{cases}$$

Note that there is zero chance that the serial interval is less than the sum of the two incubation periods. Also, when $t\geq30$, the effect of mosquito mortality does not apply to the portion of the serial interval attributed to the intrinsic incubation period.

As noted above, we treat movement independently from mosquito and disease ecology, and as such, the approximate likelihood of a causal link between two cases can be broken into it constituent parts. If we have two individuals who became infectious $t$ days apart and $x$ meters apart, the spatio-temporal transmission distribution of malaria, denoted $P(x,t)$, is given as:

$$P(x,t) = M(x,t) * G(x,t)$$

It is important to acknowledge that case data recorded are not in fact the times of onset of infectiousness within the hosts, and there is no guarantee that the time between two sequential cases being identified corresponds identically with the time between onset of infectiousness between the two individuals. To account for the non-negligible difference between the difference between two detection times and the serial interval, we add temporal noise to $t$. specifically, for a given level of temporal noise, denoted $\sigma_t$, we convolve $P$ with Normally distributed noise. Thus, the spatio-temporal transmission distribution that we analyze the data with, denoted $K(x,t)$, is (for a given diffusion constant $D$ and temporal noise coefficient $\sigma_t$)

$$K(x,t) = \int_{-\infty}^{\infty} P(x,t+\varepsilon_t) \frac{1}{\sqrt{2\pi\sigma_t{}^2}} e^{-\frac{\varepsilon_t{}^2}{2\sigma_t{}^2}} d\varepsilon_t$$

Thus, with the exception of $D$ and $\sigma_t$, the entire spatio-temporal distribution of transmission is specified. Since both the movements of mosquitoes and humans is encapsulated by $D$, it is unclear exactly what its value should be. However, as will be discussed below, by sweeping across a variety of values, we can actually use the uncertainty in $D$ to understand which links are relatively strong and which are relatively tenuous. Similarly, the true difference between onset and detection is unknown and we will likewise sweep across potential values of $\sigma_t$.

## Identifying most likely chains of transmission

The model described above computes the likelihood that a mosquito infected by a putative index case at a particular point in space and time later infected an individual identified as a locally acquired case at a different point in space and time. The assumption that movement can be approximated by diffusion (and the use within the 'likelihood' of the exact locations where the two cases were identified) will necessarily force the computed chance of causal infection to be extremely low. As such, any probability computed using the above spatio-temporal distribution would often yield inappropriately low values—if only because every particular place and time was unlikely.

What is more useful, given the uncertainty, is to compare likelihood measures. All locally acquired cases can be assigned to a "parent" that gives the highest likelihood, even though it may be only marginally more likely than some other link. We assume due to the extremely low transmission intensity of malaria in Swaziland that each infection was only caused by a single parent and superinfection did not occur. A threshold value was chosen and tested to help identify cases that were "orphans," or unlikely to be associated with any other identified case. The most likely parent or orphan status was computed for each combination of a mesh on the constants (i.e., $D$, $\sigma_t$ and a threshold value). The aggregate data from all these assessments was used to estimate the strength of some particular connection. Links that only occur in the rarest of cases (i.e. when the diffusion constant is extremely high) are tenuous.

For the Swaziland data, we compute the set of potential links for 400 different scenarios corresponding to 20 different values of the diffusion constant and 20 different values of $\sigma_t$. We vary $\sqrt{D}$ in even steps from 40 m to 1000 m and vary $\sigma_t$ from 0 to 19 days. For display purposes we will indicate the link that occurred the most times, although in most cases the link that occurred the most frequently was the only potential link identified for a given case. Further, we vary the relative probability threshold from 1% to 50%, also investigating the intermediary value of 10%.

## Modeling malaria receptivity

The number of direct offspring at each case location was explained by a set of spatial covariates, which described weather, geography, population density, and urbanicity (*Figure 6*). Elevation and topography have been demonstrated to influence risk through their effects on temperature and suitability for mosquito breeding (*Cohen, 2008*). The topographic wetness index (TWI), a measure

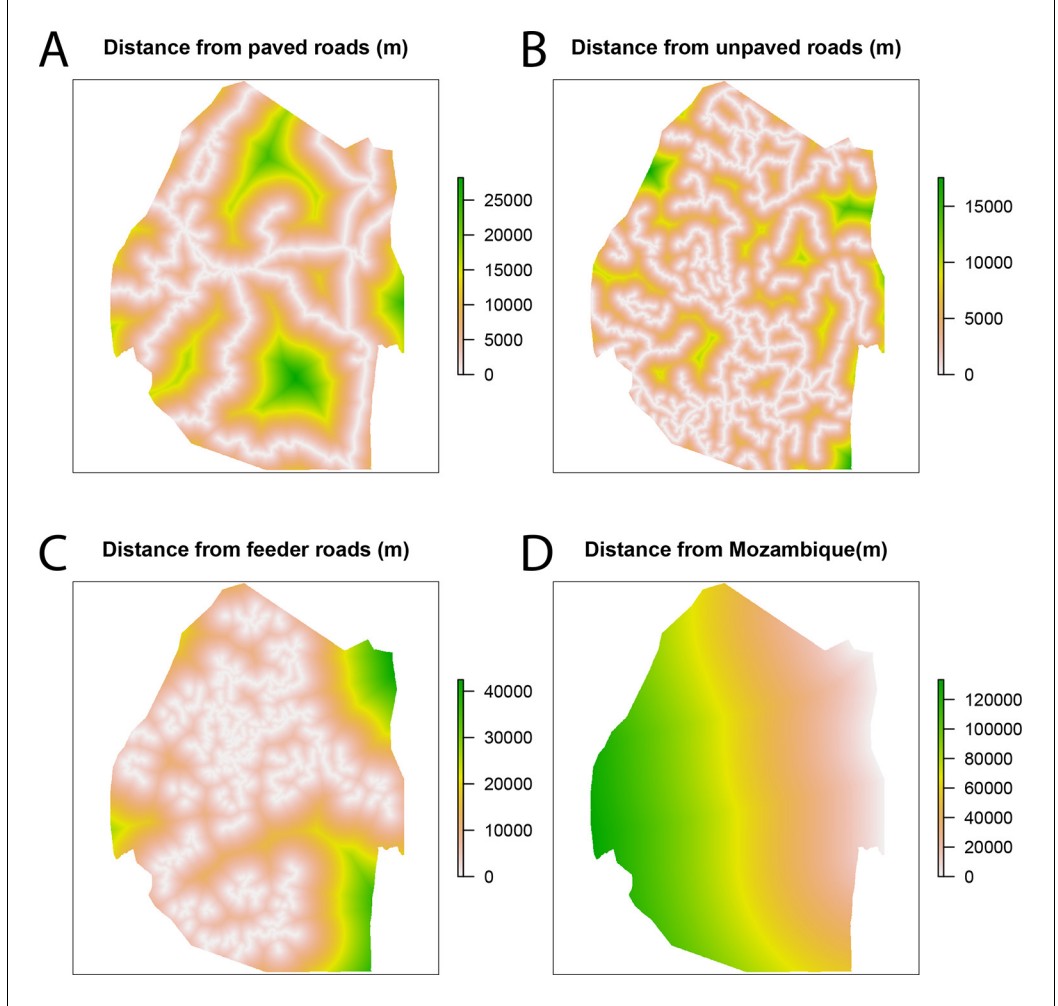

**Figure 6.** Spatial covariates for malaria receptivity regression. The four significant covariates for the malaria receptivity regression were (**A**) distance from paved roads, (**B**) distance from unpaved roads, (**C**) distance from feeder roads, and (**D**) distance from Mozambique. All distances were in meters.

**Table 1.** Zero-inflated negative binomial regression summary.

| Factor (source) | Count model coefficient | Zero-inflated coefficient |
|---|---|---|
| Intercept | 7.686407 | 1642.7199 |
| Elevation (m) (http://www.worldclim.org/bioclim) | −0.0026 | −0.65197 |
| Population (http://www.worldpop.org.uk/) | −0.017571 | −0.01190 |
| Annual Mean Temperature (0.1°C) (http://www.worldclim.org/bioclim) | 0.141979 | 30.70232 |
| Max Temperature of Warmest Month (0.1°C) (http://www.worldclim.org/bioclim) | −0.113297 | −23.66837 |
| Min Temperature of Coldest Month (0.1°C) | −0.029091 | −10.04161 |
| Precipitation of Wettest Month (mm) (http://www.worldclim.org/bioclim) | 0.008032 | 0.50592 |
| Precipitation of Driest Month (mm) (http://www.worldclim.org/bioclim) | −0.108767 | 12.04175 |
| TWI | −0.024820 | −4.02392 |
| NDVI (https://landsat.usgs.gov/) | 2.461314 | −159.39390 |
| EVI (https://landsat.usgs.gov/) | −3.732795 | 82.72595 |
| Log(theta) | −0.613861 | NA |

representing the amount of water that should enter a given spatial unit divided by the rate at which the water should flow out of that unit, was calculated from elevation as a measure for suitability for mosquito breeding habitat (*Cohen, 2008*; *Cohen, 2010a*;*2010b*). Suitability for mosquito habitat was also described using remotely sensed imagery (*Hay et al., 1998*). The normalized difference vegetation index (NDVI) (*Rouse Jr et al., 1974*) and enhanced vegetation Index (EVI) were calculated from averaged Landsat Enhanced Thematic Mapper (ETM) images from 2010 till 2013 with spatial resolution 100 m. Densely populated areas may face substantially different malaria risks from very sparsely populated, rural areas (*Hay et al., 2005*).

We used spatial zero-inflated negative binomial regression model to extrapolate the number of direct offspring from the cases locations to all points across Swaziland, producing a map of malaria receptivity at 100-meter resolution. For model selection purposes, due to the small total number of covariates for the zero-inflated negative binomial regression (12), we assessed the model fit through AIC for every sub-model (4095 models) and selected the one with the best AIC. The resulting model

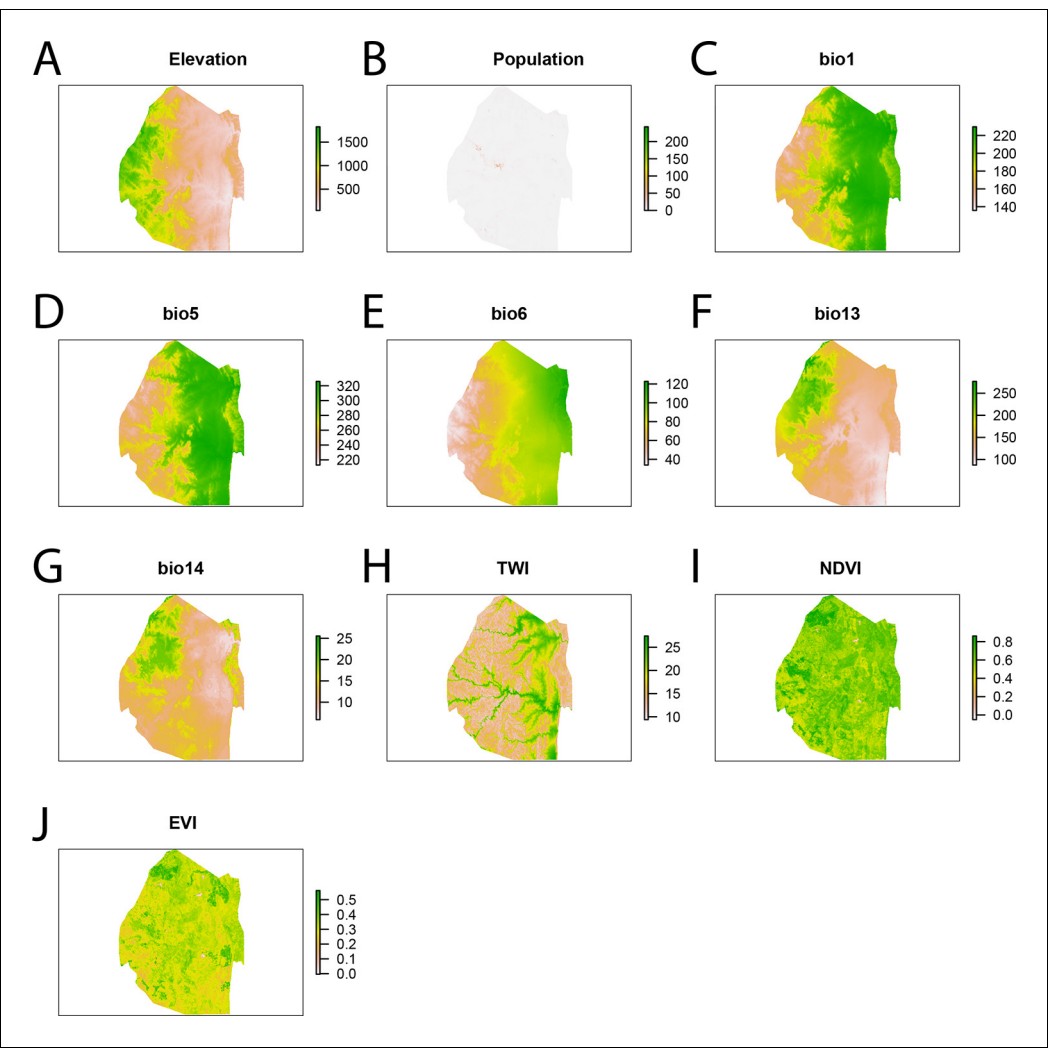

**Figure 7.** Spatial covariates for malaria importation regression. The ten significant covariates for the malaria importation regression were (**A**) elevation, (**B**) population, (**C**) annual mean temperature (bio1 - http://www. worldclim.org/bioclim), (**D**) maximum temperature of the warmest month (bio5 - http://www.worldclim.org/ bioclim), (**E**) minimum temperature of coldest month (bio6 - http://www.worldclim.org/bioclim), (**F**) precipitation of the wettest month (bio13 - http://www.worldclim.org/bioclim), (**G**) precipitation of driest month (bio14 - http:// www.worldclim.org/bioclim), (**H**) TWI, (**I**) normalized difference vegetation index, and (**J**) enhanced vegetation index.

**Table 2.** GAM logistic regression summary.

| Factor | edf | Chi.sq | p-value |
|---|---|---|---|
| Population (http://www.worldpop.org.uk/) | 6.729 | 688.01 | <2e-16 |
| Paved roads (source: country) | 5.909 | 172.49 | <2e-16 |
| Unpaved roads | 1.002 | 15.88 | 6.88e-5 |
| Feeders roads | 6.499 | 50.37 | 3e-8 |
| Distance to Mozambique (http://www.fao.org/geonetwork/srv/en/main.home) | 7.516 | 75.27 | 1.04e-12 |

(*Table 1*) retained covariates that were not found to be significant, but since we were not interested in the impact of any given covariate, but rather interpolating the observed $R_C$ values across Swaziland, backwards selection would have been inappropriate. All analysis was conducted using R, version 3.1.1 (*Team RDC, 2011*). It is important to note here that there are numerous other models that are 'almost' as good as the best. The regression algorithm is a middle step between linking cases together and making operational recommendations. We have not reported all AIC values, and there are other models that give similar AIC values and result in similar maps of receptivity. The particular parameters chosen (and their particular coefficients) are only a step of our algorithm, which would be rerun at any time-point in the future given new data and which would likely result in a different "best" model linking spatial covariates to the output of the first part of our algorithm.

## Modeling malaria importation

The risk of importing malaria from endemic countries to Swaziland is assumed to be a function of population density, distance to Mozambique and distance to roads (*Figure 7*). Values for each of the covariates were compared between the locations of the households of patients identified with imported acquired infections and randomly selected "background" points from across Swaziland. Background points do not necessarily indicate the absence of transmission, but instead characterize the environment of the country (*Anderson, 2006*). A sample of 10,000 background points (*Anderson, 2006*; *Phillips and Dudík, 2008*) was selected randomly across Swaziland. The observed importation points as well as the 10,000 background points were combined in a GAM logistic regression (*Table 2*, *Figure 7*). GAMs were implemented using the 'mgcv' package in R (*Wood, 2011*) and fit by maximizing the restricted maximum likelihood to reduce bias and over-fitting of the smooth splines.

## Malariogenic potential

The relevant concept for malaria transmission in elimination setting was named "vulnerability" by the World Health Organization (WHO) and defined qualitatively as "*the frequent influx of infected individuals or groups and/or infective anophelines*"; quantitatively, the rate of malaria importation includes all parasites that cross the border in humans and vectors. The impact of vulnerability depends on an area's "receptivity" to malaria which reflects the conditions of transmission "*through the abundant presence of vector anophelines and the existence of other ecological and climatic factors*". Receptivity is defined quantitatively as the effective reproduction number $R_C$, which describes the expected number of secondary human infections originating from a single, untreated infected human taking into account vector control measures.

The product of the receptivity and vulnerability was named malariogenic potential.

## Additional information

### Funding

| Funder | Grant reference number | Author |
|---|---|---|
| Bill and Melinda Gates Foundation | OPP1110495 | Robert C Reiner Jr<br>T Alex Perkins<br>David L Smith |

| Funder | Grant reference | Author |
|---|---|---|
| The Research and Policy for Infectious Disease Dynamics (RAPIDD) program of the Science and Technology Directorate, Department of Homeland Security, and the Fogarty International Center, National Institutes of Health | | Robert C Reiner Jr<br>T Alex Perkins<br>Andrew J Tatem<br>David L Smith |
| Bill and Melinda Gates Foundation | OPP1109772 | Arnaud Le Menach<br>Justin M Cohen |
| Swaziland Ministry of Health | | Michelle S Hsiang |
| National Institute of Allergy and Infectious Diseases | 7K23AI101012 | Michelle S Hsiang |
| Burroughs Wellcome Fund | SPA0000798 | Michelle S Hsiang |
| Bill and Melinda Gates Foundation | OPP1013170 | Michelle S Hsiang |
| Bill and Melinda Gates Foundation | OPP1132226 | T Alex Perkins<br>Bryan Greenhouse |
| National Institutes of Health | ICMER U19 AI089674 | Bryan Greenhouse<br>David L Smith |
| Bill and Melinda Gates Foundation | OPP1106427 | Andrew J Tatem |
| Bill and Melinda Gates Foundation | OPP1032350 | Andrew J Tatem |
| Wellcome Trust | | David L Smith |

The funders had no role in study design, data collection and interpretation, or the decision to submit the work for publication.

## Author contributions

RCR, Conception and design, Analysis and interpretation of data, Drafting or revising the article; ALM, DLS, Conception and design, Acquisition of data, Analysis and interpretation of data, Drafting or revising the article; SK, NN, MSH, Acquisition of data, Drafting or revising the article; TAP, BG, AJT, Analysis and interpretation of data, Drafting or revising the article; JMC, Conception and design, Acquisition of data, Drafting or revising the article

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
