## [Decision Letter]

Thank you for submitting your work entitled "Mapping residual transmission for malaria elimination" for peer review at *eLife*. Your submission has been favorably evaluated by Prabhat Jha (Senior editor), Mark Jit (Reviewing editor), and two reviewers, one of whom, John Drake, has agreed to reveal his identity.

The reviewers have discussed the reviews with one another and the Reviewing editor has drafted this decision to help you prepare a revised submission.

The reviewers and the Reviewing editor agreed that your paper takes a unique approach to modelling malaria transmission – that of linking cases with likely transmission pairs or groups. While similar approaches have been taken with viral respiratory illness (and likely other pathogens that we are not aware of) this appears to be the first application to a vector borne disease. It is sophisticated but not overly complicated and represents an important benchmark on the path to developing better analytics to guide and document the success of malaria elimination programs in sub-Saharan Africa. The analysis appears sound and internally consistent.

There are a few issues that we would like you to address, listed below. In addition, we normally require that modelling papers in *eLife* make their data openly accessible. Please could you tell us of your arrangements for this, or otherwise justify why you are not able to do so.

1) Can any indication of the temporal stability of the map of *R_c_* be given? It is essential to know if the apparently smoldering foci are stable and therefore targets of intervention, or variable and therefore blanket control measures are all that is tractable.

2) We are not clear what view you have of the number of undetected cases or what effect that might have on their map. Presumably it is theoretically possible that all the *R_c_*s are, in fact, several fold higher and case detection frustrates the exercise?

3) How much transmission might be asymptomatic? Swaziland was previously at higher transmission intensity and therefore many adults may be able to tolerate parasitaemia without becoming unwell.

4) When results on pairs of cases are presented in one figure then probabilities are assigned. However elsewhere and in the text the implication is made that a dichotomy has been made between cases that are linked and single cases – how have you moved from continuous probability to a dichotomy?

5) Genetic testing would be a more definitive way of distinguishing linked from unlinked cases. Can you comment on the likely utility of their modelling approach once genetic testing is available?

6) Is the "smoldering transmission" reflective of long-term transmission in the absence of imported cases, or does it require an imported case every so often to keep it smoldering?

7) Fine-grained needs a definition (pixel size? Or is it a point-process with pixelation simply for presentation purposes?).

8) There is a confluent area of high *R_c_* in the North East. In the rest of the country there are occasional dots of high *R_c_* scattered around. What is the possibility that these would have arisen by chance? The model implies very multiple comparisons. If a simulation was carried out where the episodes were randomly distributed through the population then how often would one see a "spike" of red colour in the map just due to chance?

9) If immigrants tend to go to the same places and at the same time (e.g. because they are all being recruited to work in the same factory) then that would increase the likelihood that the two cases would spuriously be associated in time and place despite not having a transmission cycle. Would that source of spurious linkage be accounted for in the migration data that was used?

10) Is there an issue with spatial autocorrelation (Figure 2 and zero inflated negative binomial regression)? Were the errors inspected? (In general, regression diagnostics, including differences in AIC values among models, have not been reported.)

11) What are the units in Table 1 and Table 2? Were data centered and scaled prior to fitting GAMs? In my experience, this sometimes improves performance.

---

## [Author Response]

*[…] There are a few issues that we would like you to address, listed below. In addition, we normally require that modelling papers in* eLife *make their data openly accessible. Please could you tell us of your arrangements for this, or otherwise justify why you are not able to do so.*

*1) Can any indication of the temporal stability of the map of* R_c_
*be given? It is essential to know if the apparently smoldering foci are stable and therefore targets of intervention, or variable and therefore blanket control measures are all that is tractable.*

We have repeated our analysis using the first and second halves of the data (before and after July 1, 2012). There are some differences in the *R_c_* maps, but there are also many similarities. One difficulty in interpreting the “stability” of our estimated *R_c_* maps in this low-transmission setting is that in practice *R_c_* will be based on observed cases. We consider the *R_c_* map analogous to a map of where fires may occur if a match was dropped. If you only have a few matches dropped across the landscape, it will be difficult to know which areas are most flammable. We have added these follow-up analyses and some discussion of the results to the manuscript in the “Malariogenic Potential” section.

One interesting result of the stability analysis was that in both halves of the data, the max *R_c_* by pixel was considerably higher (35.36 and 3.13 in the first and second half, respectively) as well as the percent of the population living in the highest *R_c_* group.

*2) We are not clear what view you have of the number of undetected cases or what effect that might have on their map. Presumably it is theoretically possible that all the* R_c_s *are, in fact, several fold higher and case detection frustrates the exercise?*

We have added text at the end of the “Occult Local transmission” section to further discuss the impact of undetected cases on our analyses. It is unclear that missing cases would increase *R_c_*s. For example, consider a case (say case A) that appeared to “cause” 2 other cases (say, case B in their village 30 days later and case C in a different village a few km away 45 days later). Case A’s *R_c_* would be estimated as 2. Now, if there was a missing case (say case D) in that other village that occurred 30 days before case C (15 days after case A), then we would say that A “caused” B, D “caused” C and both *R_c_*s (A’s and D’s) would be 1. Missing offspring would result in increased *R_c_*s but missing parents would result in decreased *R_c_*.

*3) How much transmission might be asymptomatic? Swaziland was previously at higher transmission intensity and therefore many adults may be able to tolerate parasitaemia without becoming unwell.*

In response to the previous reviewers’ comment, we added text discussing asymptomatic. One part of this text directly addresses this question. In several circumstances active case detection was conducted around a case to assess the level of transmission and found almost no additional infections (53/7307 in 2014-2015). While this was not done universally, it is consistent with the assumption that there is not a large pool of unreported infections (asymptomatic or otherwise) for every case.

*4) When results on pairs of cases are presented in one figure then probabilities are assigned. However elsewhere and in the text the implication is made that a dichotomy has been made between cases that are linked and single cases – how have you moved from continuous probability to a dichotomy?*

We used a threshold to differentiate between unlikely links and spurious associations due to that fact that some case has to be “most likely” even when none are plausible (see “Identifying most likely chains of transmission” section of the Methods, paragraph two, sentence four. We have added text on lines within the main text emphasizing this in the “Mapping Receptivity” section).

*5) Genetic testing would be a more definitive way of distinguishing linked from unlinked cases. Can you comment on the likely utility of their modelling approach once genetic testing is available?*

We have added to our discussion on this topic in the last paragraph of the Discussion to reinforce our belief that even as genetic testing methods improve, our approach will be useful as a baseline.

*6) Is the "smoldering transmission" reflective of long-term transmission in the absence of imported cases, or does it require an imported case every so often to keep it smoldering?*

Once a case exists, it enters the algorithm in the exact same way if it is an “imported” case or a “local” case. As such, our model wouldn’t predict that there is need for “new blood” (or a “new strain”). We have added text in the “Mapping Receptivity” section to clarify this point.

7) Fine-grained needs a definition (pixel size? Or is it a point-process with pixelation simply for presentation purposes?).

We have added text at the end of the “Residual Transmission and Elimination” section to clarify. Our algorithm (the case matching algorithm) occurs in continuous space. Our spatial regression that converts observed *R_c_*s into a smoothed *R_c_* map, as well as all other smoothed maps, is a pixel-driven process. Arguably, it too could be conducted at any resolution for which covariate layers exist.

*8) There is a confluent area of high* R_c_
*in the North East. In the rest of the country there are occasional dots of high* R_c_
*scattered around. What is the possibility that these would have arisen by chance? The model implies very multiple comparisons. If a simulation was carried out where the episodes were randomly distributed through the population then how often would one see a "spike" of red colour in the map just due to chance?*

This is an excellent question. We are following up these analyses with a statistical analysis of our algorithm using bootstrapping. We have added text in the “Mapping Receptivity” section indicating that assessing how likely it would be to see a “spike” of red due to chance is important to appropriately interpreting these maps for the purpose of resource allocation (resource allocation is the main topic of the follow-up analysis and assessing the signal/noise ratio of these maps is key for that endeavor).

This is also tangentially discussed in the “Modeling malaria receptivity” section of the Methods in response to the second half of Comment 10 (see below). A key distinction we now emphasize further is that we are presenting an algorithm to direct operational decisions. The "model selection" procedure for smoothing the calculated *R_c_* values into a map consisted of the comparison of 2^12^-1=4095 models. This approach is most useful when fit using recent data to understand the transmission landscape at that time point.

*9) If immigrants tend to go to the same places and at the same time (e.g. because they are all being recruited to work in the same factory) then that would increase the likelihood that the two cases would spuriously be associated in time and place despite not having a transmission cycle. Would that source of spurious linkage be accounted for in the migration data that was used?*

The immigration process you describe would certainly influence the related analysis of “which areas are seeded by which countries”. As far as our analyses are concerned, both of the immigrants you describe would be treated identically. However, they would not link to each other since they would both be considered “imported” cases and imported cases are assumed by our approach to never be caused by other cases. Since they would both be infectious at, for example, the same factory, then they should both “count” the same in terms of our approach.

*10) Is there an issue with spatial autocorrelation (Figure 2 and zero inflated negative binomial regression)? Were the errors inspected? (In general, regression diagnostics, including differences in AIC values among models, have not been reported.)*

Given the extremely low numbers of cases, accounting for spatial autocorrelation is quite difficult and will be further investigated in our follow-up analysis where we assess the concerns brought up in Comment 8.

In response to the second half of this comment, we have added most of the desired details, but emphasized that the exact values found in the “best” model are not the focus of our analysis. Using AIC, we compared 2^12^-1=4095 models. There are numerous other models that are ‘almost’ as good as the best. We have emphasized in the “Modeling malaria receptivity” section of the Methods that our regression algorithm is a middle step between linking cases together and making operational recommendations. We do not think it would be useful to report the 4095 AIC values. Our analysis is about the entire algorithm, which would be rerun at any time-point in the future given new data and which would likely result in a different “best” model linking spatial covariates to the output of the first part of our algorithm.

*11) What are the units in Table 1 and Table 2? Were data centered and scaled prior to fitting GAMs? In my experience, this sometimes improves performance.*

We have added units to Table 1 for the factors that have units. Table 2 does not display any values that have units. We centered and scaled the data and found the same results. Not only were the effective degrees of freedom essentially identical, but so were the fitted smooths. We have not included this analysis in the manuscript, as we did not think it was critical, but could do so if required.